# Neural network models for sequence-based TCR and HLA association prediction

Si Liu[1]\*, Philip Bradley[2,3], Wei Sun[1,4,5]\*

**1** Biostatistics Program, Public Health Sciences Division, Fred Hutchinson Cancer Center, Seattle, Washington, United States of America, **2** Herbold Computational Biology Program, Public Health Sciences Division, Fred Hutchinson Cancer Center, Seattle, Washington, United States of America, **3** Institute for Protein Design. University of Washington, Seattle, Washington, United States of America, **4** Department of Biostatistics, University of Washington, Seattle, Washington, United States of America, **5** Department of Biostatistics, University of North Carolina, Chapel Hill, North Carolina, United States of America

\* sliu3@fredhutch.org (SL); wsun@fredhutch.org (WS)

## Abstract

T cells rely on their T cell receptors (TCRs) to discern foreign antigens presented by human leukocyte antigen (HLA) proteins. The TCRs of an individual contain a record of this individual's past immune activities, such as immune response to infections or vaccines. Mining the TCR data may recover useful information or biomarkers for immune related diseases or conditions. Some TCRs are observed only in the individuals with certain HLA alleles, and thus characterizing TCRs requires a thorough understanding of TCR-HLA associations. The extensive diversity of HLA alleles and the rareness of some HLA alleles present a formidable challenge for this task. Existing methods either treat HLA as a categorical variable or represent an HLA by its alphanumeric name, and have limited ability to generalize to the HLAs that are not seen in the training process. To address this challenge, we propose a neural network-based method named Deep learning Prediction of TCR-HLA association (DePTH) to predict TCR-HLA associations based on their amino acid sequences. We demonstrate that DePTH is capable of making reasonable predictions for TCR-HLA associations, even when neither the HLA nor the TCR have been included in the training dataset. Furthermore, we establish that DePTH can be used to quantify the functional similarities among HLA alleles, and that these HLA similarities are associated with the survival outcomes of cancer patients who received immune checkpoint blockade treatments.

## Author summary

T cells are critical components of the human immune system. A T cell receptor (TCR) is a protein complex found on the surface of T cells, and it determines the antigens that a T cell recognizes. The TCR repertoire, which is the collection of all the TCRs within an individual, provides rich information about the history and current activities of the immune system. A TCR-antigen interaction is mediated by a protein called human leukocyte antigen (HLA). HLA genes are highly polymorphic in the human population, contributing to the variation of human immune response to different types of antigens. Studying the

**Data Availability Statement:** The TCR repertoire data of 666 individuals, which were generated by Emerson et al., were downloaded from https://clients.adaptivebiotech.com/pub/Emerson-2017-

NatGen in 2021, and the HLA data file for these individuals, which was generated by DeWitt et al., was downloaded from Zenodo database (doi:10.5281/zenodo.1248193). The file with survival outcome and HLA-I information for 1,535 advanced cancer patients treated with ICB, which was compiled by Chowell et al. [20], was downloaded from www.sciencemag.org/content/359/6375/582/suppl/DC1. The whole exome sequencing data and the file with survival outcome for the 144 patients with melanoma and treated with PD1 ICB [24], were downloaded NCBI SRA, and the relevant files were located by dbGAP accession number phs000452.v3.p1 (https://www.ncbi.nlm.nih.gov/projects/gap/cgi-bin/study.cgi?study_id=phs000452.v2.p1). For the TCGA cohort [25], the table of HLA information [26] is downloaded from https://gdc.cancer.gov/about-data/publications/panimmune, the table of somatic mutation information [27] is downloaded from https://gdc.cancer.gov/about-data/publications/mc3-2017, and the table of survival outcomes [28] is downloaded from https://gdc.cancer.gov/about-data/publications/pancanatlas. The CLAIRE software, which was provided by Glazer et al. [11], is also available at https://github.com/louzounlab/CLAIRE. Our data analysis pipeline is available at https://github.com/Sun-lab/DePTH_pipeline. Our software package is available at https://github.com/Sun-lab/DePTH and also uploaded to the Python Package Index (PyPI). Both repositories are licensed under the open source MIT License. A tutorial on package installation and usage is available at https://liusi2019.github.io/DePTH-tutorial/.

**Funding:** SL PB and WS were supported in part by R56 AI169192 from National Institute of Allergy and Infectious Diseases. PB was also supported in part R35GM141457. The funders had no role in study design, data collection and analysis, decision to publish, or preparation of the manuscript.

**Competing interests:** None.

associations between TCRs and HLAs can help us identify functional TCRs given the HLAs of an individual. To this end, we develop a deep learning method to predict TCR-HLA associations based on their amino acid sequences. This method allows us to borrow information across different HLA genes. We demonstrate that the predictions of our model can be used to quantify the functional similarities of HLA alleles and such similarities are associated with cancer patients' survival outcome.

## Introduction

T cells play a critical role in the human immune response by recognizing antigens presented on cell surfaces by human leukocyte antigens (HLAs). This specificity of a T cell's ability to recognize an antigen is dictated by its T-cell receptor (TCR), a protein complex found on the surface of T cells. During an immune response, T cells that recognize foreign antigens undergo clonal expansion, resulting in one or multiple clones of T cells with identical TCRs within each clone. Once the cells presenting the foreign antigens are cleared, the expanded clones contract to steady memory states and their TCRs can be detected years after the initial infection. A healthy person has a repertoire of tens of millions of TCRs [1]. Most of these TCRs are rare in the human population (or even private to an individual) because they are generated by a stochastic process. However, some TCRs are shared across individuals and they are known as public TCRs. These public TCRs likely arise because they are from the memory T cells responding to the same antigen (e.g., flu virus) in different individuals, or because they are generated with high probability [2]. Therefore, public TCRs can be used to infer immune response, such as infection history [1, 3].

HLA is the human version of major histocompatibility complex (MHC). The genetic locus encoding the HLA proteins is the most polymorphic region in the human genome [4, 5]. There are more than 15,000 classical HLA alleles [6]. We and others have shown that many public TCRs are restricted to certain HLA alleles by examining the co-occurrence of HLAs and TCRs in a subset of individuals [1, 7]. This approach is limited because it can only detect the associations of the HLAs and TCRs that are abundant in the population. It is desirable to have a more flexible method that can predict the association for any TCR-HLA pair. Such a method can be useful for many studies of immune response. For example, it can help define the functional similarities of HLA alleles and quantify the TCR-recognition capacity of a set of HLA alleles of a human being.

The challenge to train such a model is that both TCRs and HLAs are highly diverse. Moreover, each person can have up to 16 HLA alleles: up to six for three HLA class I (HLA-I) genes (two for each of *HLA-A*, *HLA-B* and *HLA-C* genes) and up to ten for three HLA class II (HLA-II) genes (two for *HLA-DR*, four for *HLA-DP* and four for *HLA-DQ*). To address this challenge, we propose a deep learning method named "DePTH", which is short for **De**ep learning **P**rediction of **T**CR-**H**LA association. The main idea is to represent a TCR and an HLA by their amino acid sequences and learn their association by a neural network. This setup allows us to borrow information across HLAs and TCRs by exploiting their sequence similarity, and once the model is learnt, we can make a prediction for any TCR-HLA pair, as long as we know their sequences.

The TCR-HLA pairs for model training are constructed using TCR data of 666 individuals generated by Emerson et al. [1] and the corresponding HLA data reported by DeWitt et al. [7]. In later narrative, we refer to this data resource as "Emerson data". We identified associated TCR-HLA pairs by their co-occurrence pattern. Our DePTH model learns to predict whether

a TCR-HLA pair is associated or not based on their amino acid sequence information. We use all positive pairs (i.e., co-occurred TCR-HLA pairs) and sample the negative pairs (i.e., not co-occurred TCR-HLA pairs) through a procedure designed to avoid potential bias.

Some recent works study the (TCR, HLA, antigen) three-way interactions by neural networks [8–10]. We focus on a different problem to model the associations between TCRs and HLAs, without being constrained to a specific antigen. To the best of our knowledge, the only other work that focuses on predicting TCR-HLA association is CLAIRE [11], which also uses a neural network model. Our work differs from CLAIRE in two aspects. First, DePTH represents an HLA allele by its sequence while CLAIRE represents an HLA allele by its alphanumeric allele name (e.g., *HLA-A*03:02*), which cannot fully capture the similarity of the sequences of two HLA alleles. Second, the training data are different. We identify associated TCR-HLA pairs by their co-occurrence [1, 7]. In contrast, CLAIRE uses associated TCR-HLA pairs from the database McPAS [12]. We show later that a model trained on one dataset does not generalize to the other, suggesting systematic difference of the two datasets. We argue that identifying positive TCR-HLA pairs by their co-occurrence in a sample of human population is more objective and is more likely to be generalizable to other samples of human population. In addition, it enables us to include more HLAs in the training process while the positive TCR-HLA pairs in McPAS are dominated by a few HLAs. Later in the Results Section, we show that when trained on co-occurrence data, DePTH outperforms CLAIRE, and when trained on McPAS data, DePTH makes more accurate prediction for all HLA-I alleles except one HLA allele that is the most abundant HLA allele in the McPAS dataset.

## Materials and methods

### Generate TCR-HLA pairs

To collect training data, we use the co-occurrence of TCR-HLA pairs to select associated TCR-HLA pairs. Based on the first batch of 666 individuals from Emerson data [1], we treat the TCRs that appear in at least two individuals as public TCRs and assess the association between each public TCR and each HLA allele by one-sided Fisher's exact test. At p-value cutoff corresponding to FDR 0.05 (Fig 1(A)), we selected 6,423 associated TCR-HLA pairs out of 742,832,595 TCR-HLA pairs involving HLA-I alleles and 11,037 associated TCR-HLA pairs out of 1,136,096,910 TCR-HLA pairs involving HLA-II alleles. These associated pairs are viewed as positive pairs.

When choosing the negative TCR-HLA pairs from those non-positive pairs, we follow two rules to avoid possible bias. The first rule is about the frequency of HLA alleles. Some HLA alleles appear much more frequently than other HLA alleles among the positive pairs (Fig 1 (B), left panel). For example, *HLA-B*08:01* appears in 878 positive TCR-HLA pairs and *HLA-A*03:02* appears only in one. If we randomly sample negative pairs, the neural network may be biased to positive TCR-HLA pairs based on HLA allele frequency in training data. Therefore, we sample negative pairs so that the HLA frequencies in negative pairs are proportional to those among positive pairs. The second rule is to control the population frequency of TCRs. For each HLA allele, since the positive pairs are selected by Fisher's exact test, the TCRs with higher population frequency are more likely to be selected (Fig 1(B), right panel). To avoid potential bias to score more prevalent TCRs as positives, for each HLA allele, we select the TCRs to form negative pairs so that their population frequencies are comparable to those of the TCRs in positive pairs. More details on training data preparation can be found in Sections 1.1 to 1.3 of S1 Appendix.

In our training data, labels of positive and negative TCR-HLA pairs are encoded as 1 and 0, respectively. For each pair, the input for an HLA is its sequence at positions that contact the

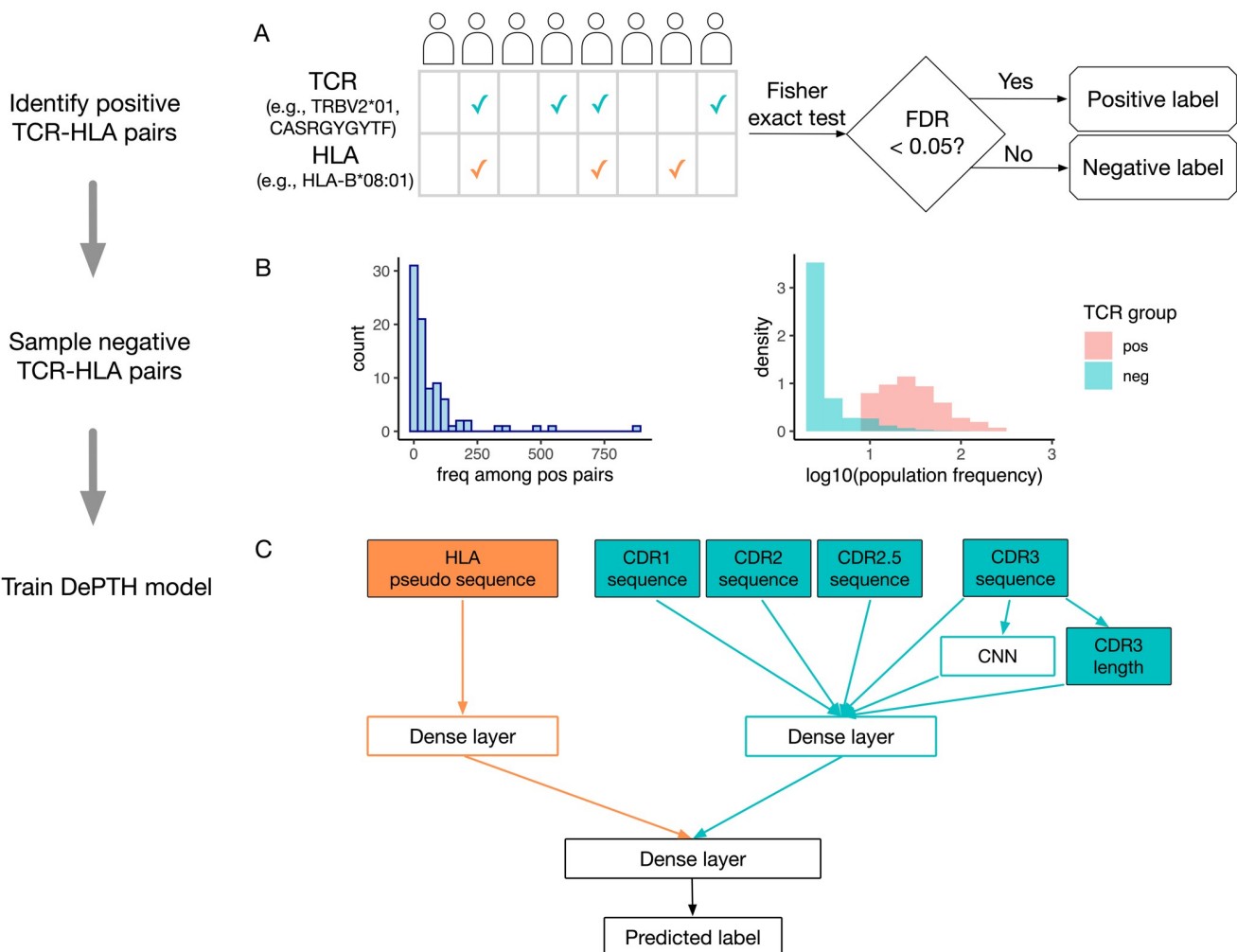

**Fig 1. Summary for data generation and model training.** (A) A toy example to illustrate the selection of positive TCR-HLA pairs by their co-occurrence. A check mark indicates the TCR or HLA is observed in an individual. The p-value for the co-occurrence of a TCR-HLA pair is computed by one-sided Fisher's exact test. To account for multiple testing across all TCR-HLA pairs, a TCR-HLA pair is considered associated if the corresponding False Discovery Rate (FDR) < 0.05. (B) The left panel is a histogram of the number of TCR-HLA pairs involving each HLA-I allele among the positive pairs identified from the Emerson data. The two overlaid histograms in the right panel show the population frequencies of TCRs that are associated with *HLA-B*01:01* or not, based on Emerson data. (C) Illustration of the components in the DePTH model. Each TCR is represented by its amino acid sequences from CDR1, CDR2, CDR2.5 and CDR3 parts, and each HLA is represented by its pseudo sequence (part of its sequence that interact with TCRs or antigens). DePTH learns to predict whether a TCR-HLA pair is associated (positive) or not (negative). For HLA part, the encoded pseudo sequence is flattened and passed to a dense layer. For TCR part, the encoded CDR3 amino acid sequence is passed into a CNN (convolutional neural network). The output of the CNN is concatenated together with CDR3 length and the encoded sequences of CDR1, CDR2, CDR2.5, and CDR3. Then they are passed to a dense layer. Finally, the outputs from TCR and HLA dense layers are concatenated and passed to one or two dense layers to make the final prediction.

antigens or TCRs, and we refer to it as HLA pseudo sequence (see Section 2.1 of S1 Appendix for details). A TCR is composed of an $\alpha$ chain and a $\beta$ chain with more diversity on $\beta$ chain. Most TCR data only include the $\beta$ chain, which is the case for the Emerson data we use. Therefore we only consider $\beta$ chain in this work, though our model can be easily extended to the situations where both $\alpha$ chain and $\beta$ chain are available. The input for a TCR beta chain is its sequence in a few complementarity-determining regions (CDRs): CDR1, CDR2, CDR2.5 and CDR3, which are known to contact either the antigen or the HLA. CDR1, CDR2, and CDR2.5 are part of the V gene of a TCR (see Section 2.2 of S1 Appendix for details). CDR3, which is

the most diverse region in a TCR, has variable lengths across TCRs. We pad all CDR3 sequences to length 27 by aligning them at both ends and padding the middle region by a character ".", following the way of padding as in [13] (see Section 3.1 of S1 Appendix for more details). The length of CDR3 before padding is included as an additional categorical input to the neural network.

## Train neural network models

Our neural network includes a dense layer to encode the HLA sequence, and a combination of a CNN (a convolutional layer plus a max-pooling layer) and a dense layer to encode the TCR sequence. CDR3 is the most diverse region in a TCR and it plays a more important role in TCR-HLA-antigen interaction, therefore we apply a CNN to CDR3 to capture non-position specific sequence motifs. The encoding of HLA and TCR are concatenated together and passed through one or two dense layers plus a dropout layer before final output is generated (Fig 1(C)).

The number of negative pairs that we sample for each HLA allele is five times of the number of positive pairs. We randomly split both positive and negative pairs into three sets: 60% for training, 20% for validation and 20% for testing. The ratio of the number of positive pairs vs. that of negative pairs is 1:5 for all three sets. We train separate neural networks to predict TCR-HLA associations involving HLA-I alleles and HLA-II alleles, respectively. For hyperparameters, we consider amino acid encoding methods, number of dense layers and dropout probabilities in a dropout layer when predicting TCR-HLA association with concatenated inputs. We run cross-validation to choose the best hyperparameter setting. Using the chosen hyperparameters, we train our model on training data, choose when to stop training based on the AUC (area under the ROC curve) on the validation data, save the model with the best validation AUC and evaluate its performance on test data. See S1 Appendix Sections 4.1 and 4.2 for details on model training and hyperparameter selection.

## Quantify HLA similarities and define HLA-based metrics for association studies

We can quantify the similarities of two HLA alleles based on their amino acid sequences, with an implicit assumption that two HLA alleles with similar sequences also have similar functions. Alternatively, we may use DePTH to predict TCR-HLA associations and use such associations to quantify the similarity of two HLAs. The rationale is as follows. If we expect that HLAs affect clinical outcomes through T cell response, then two HLAs that are associated with similar sets of TCRs should have similar clinical outcomes. Many clinical outcomes may be related to T cell response, such as cancer patients' survival time after immunotherapy. In practice, we cannot search for HLA-associated TCRs from all possible TCRs, and thus we choose the set of TCRs based on the applications. For example, in cancer studies, we can choose TCRs from T cells with cancer reactive gene expression signatures.

Given the similarity of two HLA alleles, we can define a few HLA-based metrics for association studies between a phenotype or a clinical outcome and an individual's HLA alleles. Here we define two types of metrics, between-individual HLA distances and individual-level HLA heterozygosity metrics. For ease of description, we introduce the two metrics for HLA-I alleles. It is straightforward to extend to HLA-II alleles.

**Between-individual HLA-I distances.**   A between-individual HLA-I distance characterizes the distance between the HLA-I alleles of two individuals. We consider two approaches to define the distance.

For the first approach, we define an individual-level TCR set by taking the union of all the TCRs associated with at least one HLA allele of this individual (Fig 2(A)). Given two individual-level TCR sets, denoted by $A$ and $B$, we compute their distance by the Jaccard distance:

$$\frac{|A - B| + |B - A|}{|A \cup B|}, \tag{1}$$

which is the number of TCRs belonging to only one of the two sets divided by the number of TCRs belonging to either set (Fig 2(B)). We refer to this distance as "dist_DePTH_breadth" since it is based on prediction scores from DePTH model and the breadth of TCR set.

For the second approach, we first quantify the distance between each HLA allele from one individual and each HLA allele from the other individual, hence generating a distance matrix. Since each individual has 6 HLA-I alleles, it produces a 6 × 6 matrix. Here two homozygous HLA alleles are treated as separate entities. Then we reduce this distance matrix to a number by optimal transport [14]. Specifically, to define distance "dist_DePTH_cor", we first compute the between-allele distance as 1-Spearman's correlation of predicted TCR-HLA association scores. Then the resulting 6 × 6 matrix quantifies the cost to transfer between any two HLA alleles of the two individuals. The optimal transport method reduces this matrix to one number: the minimal cost to change all 6 HLA-I alleles from the first individual to the 6 HLA-I alleles of the second individual. The resulting minimal cost is taken as the between-individual distance.

**Individual-level HLA-I heterozygosity metrics.** Besides the between-individual distances, we also define individual-level HLA-I heterozygosity metrics to quantify the "variation" of HLA-I alleles within an individual. There are also two approaches to define the metric.

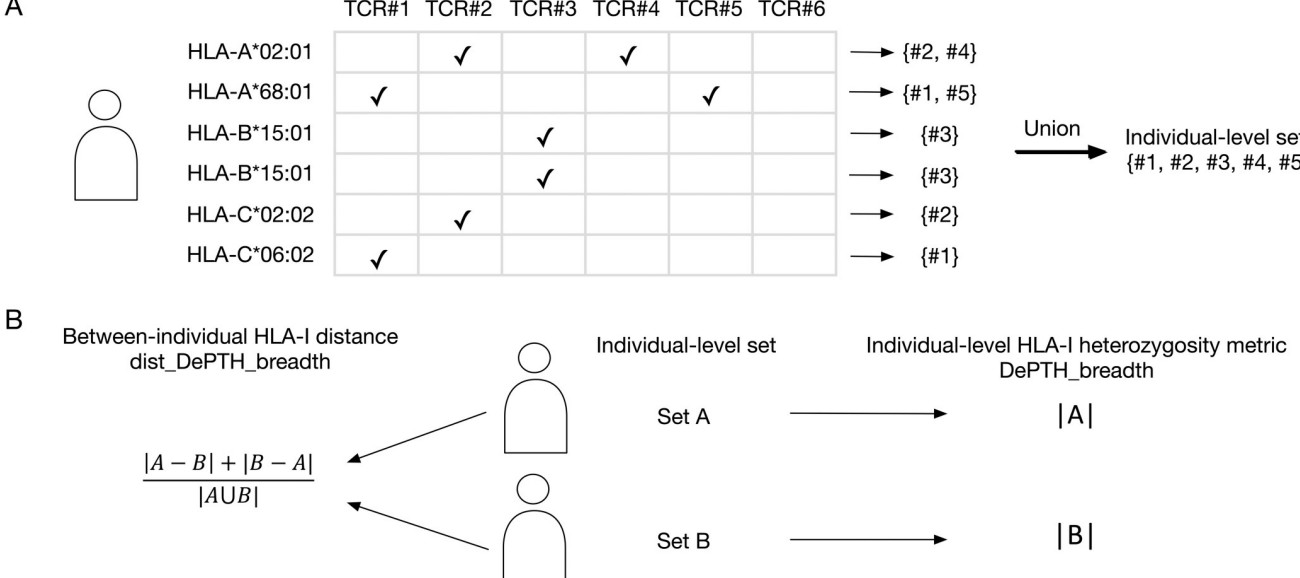

**Fig 2. A toy example to illustrate the calculation of "dist_DePTH_breadth" and "DePTH_breadth".** (A) A demonstration for the selection of the TCRs that are associated with at least one of an individual's HLAs. In the HLA vs. TCR table, a check mark indicates predicted association between an HLA (each row) and a TCR (each column). For each HLA, we collect the set of TCRs that are predicted to be associated with the HLA. The individual-level TCR set is computed by taking the union of these six TCR sets. (B) An illustration for the calculation of between-individual distance ("dist_DePTH_breadth") and individual-level HLA-I heterogeneity metric ("DePTH_breadth"). $|A|$ and $|B|$ are the sizes of the sets $A$ and $B$.

One approach, namely "DePTH_breadth", is the total number of unique TCRs that are associated with at least one HLA allele of an individual (Fig 2(B)).

The other approach has two steps. First, calculate the distance between two alleles of each HLA-I gene (i.e., *HLA-A*, *HLA-B* or *HLA-C* gene). Next, take average across three HLA-I genes as the individual-level HLA-I heterozygosity metric. Following this approach, we define two metrics "mean_DePTH_cor" and "mean_DePTH_set" where "mean_DePTH_cor" calculates the between-allele distance by 1 minus the Spearman's correlation and "mean_DePTH_set" calculates between-allele distance by the Jaccard distance.

## Results

### DePTH models achieve AUC around 0.8 on test TCR-HLA pairs

The best hyperparameter setting selected by cross-validation for HLA-I and HLA-II are the same. See Table A in S1 Appendix for the optimal hyperparameters, and Fig A in S1 Appendix for a graph showing the neural network architecture using the optimal hyperparameters. The results are not sensitive to the hyperparameters. Most hyperparameters deliver similar and good performances.

Due to the inherent randomness of stochastic gradient descent algorithm, given the same set of hyperparameters, the predictions from different random seeds are not the same. We find that an ensemble prediction from multiple models (i.e., taking average prediction scores) delivers more robust and slightly more accurate predictions. We have evaluated the ensemble prediction from $n$ models where $n$ increases from 1 to 100. The validation AUC becomes stable when the ensemble size is larger than 20 (S1 Appendix Fig B, with more discussion for choosing ensemble size in S1 Appendix Section 4.3). In the following texts, all prediction scores and evaluation results from DePTH models are based on an ensemble of 20 models trained under 20 sets of random seeds, unless specified explicitly otherwise.

We evaluate the performance to predict TCR-HLA association by three metrics, AUC, sensitivity, and specificity. The sensitivity and specificity are computed based on a prediction score cutoff of 0.5 such that TCR-HLA pairs with prediction score greater than 0.5 are classified as associated pairs. DePTH delivers good classification accuracy, with AUC 0.82 and 0.79 for HLA-I and HLA-II alleles, respectively (see Fig 3(A) for the corresponding ROC curves). While the specificities are above 80% for both HLA-I and HLA-II alleles, the sensitivities are modest around 0.63 (S1 Appendix Table B).

As a baseline model for comparison, in the situation of Emerson data involving HLA-I alleles, we train a random forest on the training data, where the inputs to random forest are the one-hot encoded amino acid sequences and CDR3 sequence length. The number of decision trees in the random forest increases in a step size of 100 until the validation AUC has not increased for two consecutive steps. The ensemble of decision trees giving the best validation AUC during the training process is chosen. When evaluating this random forest in test data, the AUC is 0.77, lower than corresponding results from DePTH. See S1 Appendix Fig C(A) for the ROC curves of DePTH and random forest.

### DePTH can generalize to TCRs and HLAs unseen during training

In order to evaluate whether our model can generalize to TCR-HLA pairs that are not seen during model training and validation, we run three leave-one-out experiments. Each time, one of the top three most frequent HLA-I alleles that appear among the positive TCR-HLA pairs, namely *HLA-B\*08:01*, *HLA-B\*07:02* and *HLA-C\*07:01*, is left out. We chose these most frequent HLA-I alleles so that there are enough HLA-TCR pairs to obtain a reliable performance evaluation.

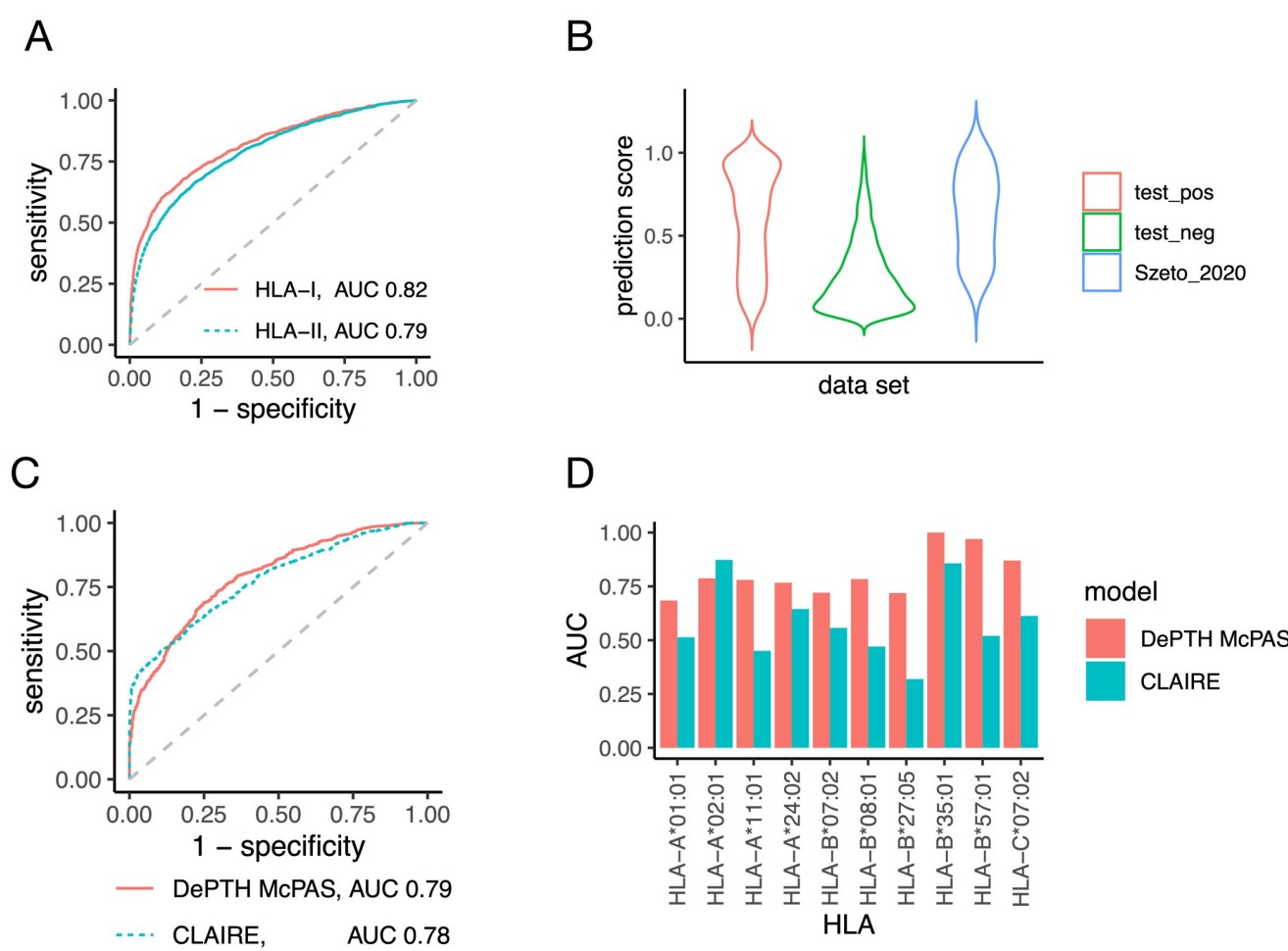

**Fig 3. The performance of DePTH models trained on Emerson data.** (A) the ROC curve of DePTH prediction on test data (part of Emerson data) for HLA-I and HLA-II, respectively. (B) Violin plots for the TCR-HLA association scores predicted by DePTH on three sets of TCR-HLA pairs: the positive pairs of Emerson test data, the negative pairs of Emerson test data, and the 54 external positive TCR-HLA pairs from solved TCR-pMHC-I structures provided by Szeto et al. [15]. The scores are given by DePTH model trained on Emerson training data. (C) and (D) compare the performance of DePTH and CLAIRE on McPAS test data while both models are trained on McPAS data. (C) ROC curves of prediction scores across all HLA-I alleles. (D) Comparison of allele-wise AUCs.

In each experiment, given the chosen HLA-I allele, for example, *HLA-B*08:01*, we construct the training, validation, and test datasets in the way such that (1) both positive and negative pairs in test data only involve *HLA-B*08:01*, while no pair in training and validation data involve *HLA-B*08:01*, and (2) there is no overlap between the TCRs from training or validation data with those from test data. To sample negative TCR-HLA pairs, we follow the same principles in terms of matching HLA frequency and TCR population frequency as those for the full data set. The ratio of the number of positive versus negative TCR-HLA pairs and the ratio of the sample size of training vs validation pairs are the same as those listed in Materials and methods section. The hyperparameter setting is chosen from cross-validation. The test AUCs from these experiments range from 0.64 to 0.69 (S1 Appendix Table C), smaller than the AUC for the full data set, but they are still much larger than 0.5, which is expected by chance. Besides, the relatively lower performance is also likely due to reduced training sample size, since none of the training/validation pairs involves any HLA or TCR from test data. For

this leave-one-out study, the optimal prediction score cutoff may be smaller than our default value of 0.5. The recall and specificity values at different prediction score cutoffs are also listed in S1 Appendix Table D for each leave-one-experiment, respectively.

## DePTH makes accurate prediction on TCR-HLA pairs from solved TCR-pMHC-I structures

We further evaluate DePTH by an independent data set provided by Szeto et al. [15], which consists of all the 81 TCR-pMHC-I complexes with solved structures deposited at Protein Data Bank by the year 2020. This data set provides a resource of TCR-HLA pairs that are known to bind together. We evaluate the performance of our model on this resource. After excluding the complexes involving non-human MHCs and additional processing (see Section 5 in S1 Appendix for more details), we end up with 54 unique TCR-HLA pairs. Based on prediction scores from our DePTH model trained on Emerson data, 67% of these TCR-HLA pairs are predicted to be positive at cutoff 0.5. When considering all cutoffs by an ROC, the AUC to distinguish these 54 TCR-HLA pairs from the negative pairs from Emerson test data is 0.87. We further examine the distributions of the prediction scores of these 54 TCR-HLA pairs, as well as the positive and negative pairs from Emerson test data (Fig 3(B)). These 54 TCR-HLA pairs have very similar prediction score distribution as the positive test set and higher prediction scores than the negative test set. This shows that DePTH model trained on Emerson data makes reasonable predictions for TCR-HLA pairs that do bind together.

We also applied the baseline random forest trained as described earlier to the same task of distinguishing the 54 Szeto 2020 TCR-HLA pairs from negative pairs in Emerson test data and obtained a resulting AUC of 0.68, which is lower than corresponding result from DePTH. See S1 Appendix Fig C(B) for the violin plots for prediction scores.

## Comparison with CLAIRE

CLAIRE is a neural network model proposed by Glazer et al. [11] in 2022 for predicting the binding of TCR and HLA. It takes as inputs the HLA allele, V$\beta$ gene and J$\beta$ genes, and TCR $\alpha$ chain information as well when available. In this section, we discuss the major differences between CLAIRE and DePTH.

**Difference in training data.**  We use the co-occurrence of HLAs and TCRs in a human population (the Emerson dataset [1]) to construct training data of associated TCR-HLA pairs. In contrast, CLAIRE mainly focuses on McPAS, which is a manually curated database [12] and relates TCRs with their associated MHCs based on published experimental data.

We show that the model trained on one dataset (either Emerson or McPAS dataset) does not generalize to the other. Specifically, we train DePTH models on one of the two datasets, and then make prediction on the other dataset. For McPAS data, the training, validation and test sets used for developing CLAIRE were obtained from the github repository for CLAIRE [11]. The process of training DePTH on the processed McPAS data is similar to that of training DePTH on Emerson data (See S1 Appendix Sections 6.1–6.2 for more details on data processing and model training).

Our model trained on processed McPAS data for HLA-I class achieves AUC 0.79 on McPAS test data, with sensitivity 0.72 and specificity 0.71. However, if we evaluate it on the Emerson test data, the AUC is only 0.52, which is close to random guess. On the other hand, if we evaluate the DePTH model trained on Emerson data by making predictions on the McPAS test data, the AUC is also 0.52. These results show that the model trained on one data source does not generalize to the other data source. This is consistent with the observation by Glazer et al. [11] that the CLAIRE model trained on McPAS data does not generalize to Emerson

data. A possible reason is the TCR-HLA pairs from McPAS data might have higher binding affinity, as conjectured by Glazer et al. [11].

**Difference in encoding the HLA and TCR input.** DePTH takes the pseudo sequence of an HLA allele (i.e., amino acid sequence at the positions that interact with an antigen or a TCR) as input. This choice allows DePTH to be applied to any HLAs with sequence information. In contrast, CLAIRE takes the name of HLA allele as input to the model, by splitting the name string into alphabetic or numerical characters and mapping each unique character to one unique integer. After this step, the HLA name string is converted into a vector of integers, which is passed as input to the next step of the model. For prediction on an HLA allele that does not appear in the training data, the generalization ability of CLAIRE will rely on how the similarity in HLA name strings affects the similarity in their binding abilities to the TCR. Furthermore, for the V gene of a TCR, DePTH takes its amino acid sequence in CDR1, CDR2 and CDR2.5 as input features, which allows generalization to previously unseen V genes. In contrast, CLAIRE treats V gene as a categorical variable.

**DePTH outperforms CLAIRE on most HLAs when trained on the same data.** To show the prediction performance difference related to the encoding of input features, we compare the DePTH trained on McPAS training data (we refer to this model as DePTH McPAS) with CLAIRE model (which was trained on McPAS data) on McPAS test data. The prediction scores from CLAIRE model are obtained by submitting the test data files to the CLAIRE model provided by Glazer et al. [11] on a web server. In terms of overall AUC based on prediction scores on all test pairs, DePTH McPAS achieves 0.79, which is slightly better than 0.78 achieved by CLAIRE model, with ROC curves shown as in Fig 3(C). Furthermore, we focus on the 10 HLA-I alleles that each appears in at least one positive TCR-HLA pair and at least one negative pair in test data. For each HLA-I allele, we compute the allele-wise AUC based on prediction scores by DePTH McPAS or CLAIRE. DePTH McPAS has higher AUC than CLAIRE in 9 out of the 10 HLA-I alleles (Fig 3(D)), except for one, *HLA-A\*02:01*, which is the most frequent HLA-I allele among the TCR-HLA pairs used for training CLAIRE and occupies 44% of TCR-HLA pairs involving HLA-I alleles. While for allele *HLA-B\*27:05*, which is a much less frequent HLA-I allele and is observed in only $\sim 1\%$ of these TCR-HLA pairs, AUCs from DePTH McPAS and CLAIRE are 0.72 and 0.32, respectively. See S1 Appendix Table E for a complete list of the frequency for each of these 10 HLA-I alleles among TCR-HLA pairs in CLAIRE training data that involve HLA-I alleles. For rare HLA alleles, DePTH has the advantage that it can learn from other HLA alleles with similar pseudo sequences.

The above results from DePTH McPAS are from an ensemble of 20 models trained with 20 sets of random seeds. Since CLAIRE model is a single model instead of an ensemble, to compare the two approaches on the level of single model, we also compare the results of one DePTH McPAS model trained with one set of random seeds with those of CLAIRE. The single DePTH McPAS model gives overall test AUC 0.76, which is lower than the 0.78 achieved by CLAIRE, with ROC curves shown in S1 Appendix Fig D(A), but the trend of allele-wise AUC comparison (S1 Appendix Fig D(B)) is consistent with the case of model ensemble, where the allele-wise AUC from the single DePTH McPAS model is higher than that from CLAIRE for all 9 HLA-I alleles, except for HLA-A\*02:01.

In addition, we have applied DePTH McPAS and CLAIRE to distinguish the 54 positive TCR-HLA pairs from the solved TCR-pMHC-I structures (i.e., Szeto 2020 data) versus negative pairs from McPAS test data. The performances show similar pattern. DePTH McPAS performs better than CLAIRE when AUCs are computed based on all TCR-HLA pairs. DePTH McPAS performs worse on the pairs only involving HLA-A\*02:01, and better on other pairs. See S1 Appendix Section 7 and Fig E for more details.

### HLA similarity metrics defined based on DePTH prediction are associated with survival outcome of cancer patients treated with Immune Checkpoint Blockade (ICB)

Given a list of TCRs relevant for a specific phenotype or clinical outcome, DePTH allows us to transform an HLA from the space of amino acid sequences (input of DePTH) to the space of TCRs, since the output of DePTH gives the association between this HLA and the given TCRs. As described in the Materials and methods section, we quantify HLA similarities based on such TCR-HLA associations, and further derive HLA metrics to evaluate distance between individuals or the degree of HLA heterozygosity within an individual. In this section, we assess the associations between different HLA metrics and survival outcome of cancer patients treated with ICB.

Chowell et al. [16] reported a dataset containing both HLA-I information and the survival outcome for 1,535 advanced cancer patients treated with ICB. From these patients, 1,443 individuals are kept for analysis after filtering and processing steps. Since the outcome is cancer related, we compile a list of TCRs from CD8+ T cells with cancer reactive signatures, using a dataset with single cell TCR and gene expression data [17]. We first identify cancer-related CD8+ T cells using known gene expression signatures [18–21], and then extract their TCRs. We apply DePTH to score the associations between each HLA-I allele from the Chowell data and the set of potentially cancer-related CD8+ TCRs. See S1 Appendix Sections 8.1 and 8.2 for more details on filtering and processing of Chowell 2018 data, extracting the list of cancer-related TCRs, and obtaining prediction scores from DePTH. We define both between-individual HLA-I distances and individual-level HLA-I heterozygosity metrics based on the prediction scores.

**Between-individual HLA-I distances.** The between-individual HLA-I distances measure how different the sets of HLA-I alleles from two individuals are. A small distance means the HLAs of the two individuals interact with similar sets of cancer reactive TCRs, which may lead to similar immune response to tumor and thus similar survival outcomes.

We compute the between-individual HLA-I distances "dist_DePTH_breadth" and "dist_DePTH_cor" using the prediction scores given by DePTH model, as described in the Materials and methods section. Using similar methods and prediction scores from CLAIRE instead of DePTH, we also create two other between-individual distances "dist_CLAIRE_breadth" and "dist_CLAIRE_cor". In addition, we define another distance "dist_AA", which is similar to "dist_DePTH_cor", except that the between-allele distance is defined based on alignment of their HLA-I pseudo sequences, following the approach by [22]. See S1 Appendix Section 9 for more details.

Based on each of these five distances, we calculate a distance matrix for all individuals. Next, we assess the association between this distance matrix and survival outcome by "MiRKAT-S", which was originally developed to assess the association between survival outcome and distance defined by microbiota [23]. we run MiRKAT-S both with and without covariates, where the covariates include age, log-transformed mutation burden, and drug type (CTLA-4, PD-1, or both). As shown in Fig 4(A), at p-value cutoff 0.05, "dist_DePTH_breadth" shows significant association with survival outcome when no covariate is included, and the significance remains when covariates are included. "dist_DePTH_cor" gives p-value around 0.10 when no covariate is included. In contrast, none of "dist_AA", "dist_CLAIRE_cor" and "dist_CLAIRE_breadth" show significance either with or without covariates.

The performance advantage over "dist_AA" indicates that our predicted scores may provide meaningful information in terms of association of HLAs with TCRs besides HLA pseudo sequences, and thus may provide better between-individual HLA-I distance metrics for

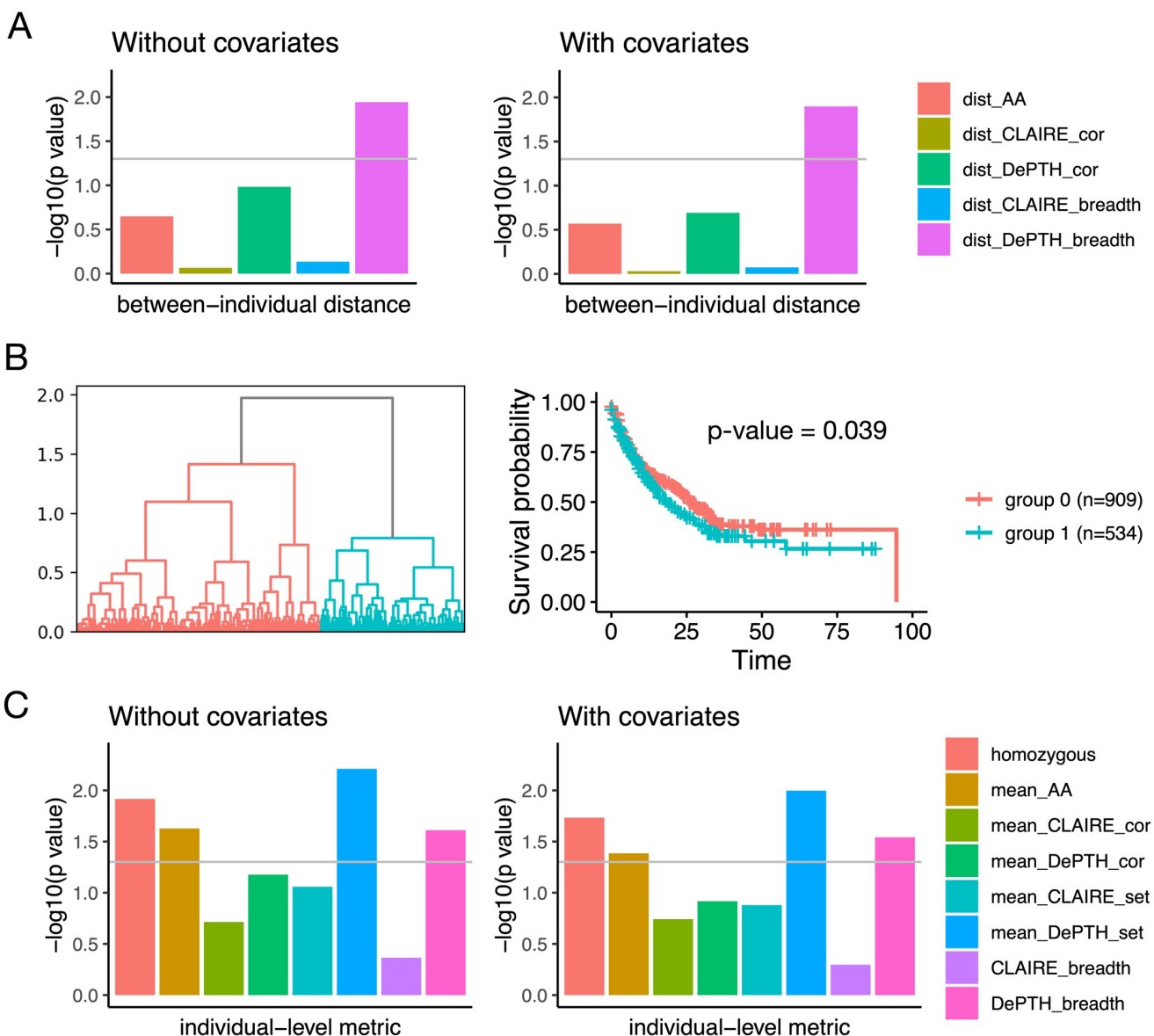

**Fig 4. Association between HLA-based metrics and survival outcomes of 1,443 cancer patients [16].** (A) and (B) summarize the results using between-individual HLA-I distance metrics. (A)-log10(p-value) of kernel regression using five different distance matrices with or without covariates. The covariates include age, log-transformed mutation burden, and drug type (CTLA-4, PD-1, or both). The grey horizontal line corresponds to p-value 0.05. (B) The dendrogram of hierarchical clustering where the distance is defined by "dist_DePTH_breadth", and the survival curves of the two subgroups identified by cutting the dendrogram tree. The p-value is computed from Cox proportional hazard regression. (C) -log10(p-value) from Cox regression for the associations between individual-level HLA-I heterozygosity metrics and survival outcome, with or without covariates. The covariates include age, log-transformed mutation burden, and drug type (CTLA-4, PD-1, or both). The grey horizontal line corresponds to p-value 0.05.

capturing the association between HLA similarity and relevant phenotype, for example, survival outcome. The advantage over "dist_CLAIRE_cor" and "dist_CLAIRE_breadth" may be due to multiple factors. One is the flexibility of our model to make meaningful predictions for rare or previous unseen HLAs. Another factor is the limited diversity of HLA-I alleles among the McPAS data used for building CLAIRE model. The McPAS data only involve 35 different HLA-I alleles, while there are 146 different HLA-I alleles from the 1,443 individuals in the Cho-well 2018 cohort. The third factor could be bias in the training data. The McPAS positive

TCR-HLA pairs that are found through experiments might tend to have stronger binding, while the Emerson positive TCR-HLA pairs got from co-occurrence pattern may have weaker binding but reveal association in more general sense, regardless of the context of specific antigens.

We also conduct an un-supervised analysis to cluster patients by hierarchical clustering using the distance matrix formed by "dist_DePTH_breadth" (Fig 4(B)). The patients clearly separate into two groups whose survival time are significantly different, with p-value 0.039 from Cox proportional hazard regression. This serves as an additional evidence that our "dist_DePTH_breadth" does capture HLA similarity that is relevant to the similarity in survival outcome.

**Individual-level HLA-I heterozygosity metrics.** The individual-level HLA-I heterozygosity metrics measure how diverse the set of HLA-I alleles within each individuals is. If an individual has more diverse HLA alleles, then we expect that her/his HLA proteins can interact with more TCRs. Thus, more diverse HLA alleles may lead to a better immune surveillance to tumor and further lead to better survival outcome.

We define eight individual-level HLA-I heterozygosity metrics to quantify the "variation" of HLA-I alleles within an individual, and assess their associations with survival outcome through cox regression. Among them, "homozygous" is a simple metric, which has value 0 if an individual has heterozygous HLA alleles for all three HLA-I genes (*HLA-A*, *HLA-B*, and *HLA-C*), and 1 otherwise. For each individual, "DePTH_breadth", "mean_DePTH_cor" and "mean_DePTH_set" are computed as described in the Materials and methods section based on the prediction scores given by DePTH, while "CLAIRE_breadth", "mean_CLAIRE_cor" and "mean_CLAIRE_set" are calculated similarly based on the prediction scores given by CLAIRE instead. In addition, we define another metric "mean_AA", which is computed in a similar way as that for "mean_DePTH_cor" and "mean_DePTH_set", but purely based on amino acids in the HLA pseudo sequences and uses the same allele-level distance as that for "dist_AA" (see S1 Appendix Section 9 for more details).

We run Cox regression on the 1,443 individuals from Chowell cohort to assess the association between each individual-level HLA-I heterozygosity metric and survival outcome. As shown in Fig 4(C), at p-value cutoff 0.05, when no covariate is included, "homozygous", "mean_AA", "mean_DePTH_set", and "DePTH_breadth" show significance and the significance remains when covariates (age, log-transformed mutation burden, and drug type) are included. These results suggest that DePTH is a useful alternative resource to quantify HLA similarities. Compared with "homozygous" which is 0/1, "mean_DePTH_set" and "DePTH_breadth" are continuous metrics and thus can provide a quantitative rather than categorical measurement.

## Discussion

We have proposed a neural network method DePTH to predict the associations between TCRs and HLAs based on their sequences. We demonstrate that two factors are important for the generalizability of DePTH: unbiased selection of training data from TCR and HLA co-occurrence patterns and representing an HLA allele by its sequence.

We demonstrate an application of DePTH to define similarities of HLA-I alleles and show the resulting HLA metrics are associated with survival outcome of cancer patients undergoing cancer immunotherapy. A closely related question is whether the HLA metrics are associated with patient response to immunotherapy. The Chowell et al. [16] dataset does not have complete treatment response information and thus we turned to another dataset of cancer patients with melanoma and treated with PD-1 blockade, with or without prior-CTLA4 treatment [24]. In the PD-1 only group, the patients with low HLA heterozygosity metrics and mutation

burdens tend to have worse outcome (S1 Appendix Figs F&G), though it is hard to evaluate the association due to small sample size. Nevertheless, these results suggest a potential direction of using HLA-I and/or HLA-II heterozygosity metrics together with mutation burden as biomarkers for patient response to immunotherapy.

In addition, to explore the relevance of our metrics in a broader range of cancer types, we also evaluated associations between HLA similarity/heterozygosity and survival outcomes in The Cancer Genome Atlas (TCGA) [25, 26] cohort for each cancer type separately. We found for certain cancer types, our metrics can identify significant associations between HLA metrics and survival outcomes that were missed when only considering whether there is HLA-I homozygosity or not. See more details on data processing and results in S1 Appendix Section 12 and S1 Appendix Tables F-I. For the Chowell 2018 cohort, all patients received ICB treatment and the survival outcomes of patients may be more closely related to the immune response and TCR-HLA interactions. While for the TCGA cohort, the treatments that the patients received can be diverse, which, together with relatively smaller number of patients under each cancer type, may have contributed to that only a few cancer types show significant associations between the HLA metrics and survival outcomes.

There are also many other situations where the results of DePTH can be very useful. For example, earlier works have shown that TCR data can be used to predict past infection (e.g., cytomegalovirus (CMV) [1] or SARS-COV-2 [3]). Since many TCRs are restricted to certain HLA alleles [1, 3, 7], incorporating HLA information could improve the prediction performance. This is a challenging task because HLAs are highly polymorphic and many HLA alleles are relatively rare in human population. Our DePTH method provides a solution to find HLA-specific TCRs and thus can help build HLA-specific TCR predictors for infection or other immune related conditions.

There are several directions that warrant further development. First, as more TCR data are accumulated, it is desirable to retrain DePTH with more data. Second, we have only considered TCR$\beta$ chain in our method due to limitation of available data. As more paired TCR$\alpha$ and $\beta$ chains are being generated (e.g., by single cell TCR data), it is possible to expand our model to include both TCR$\alpha$ and $\beta$ chains.

Finally, furthering examination to the model and pinpointing the sequential structures that are important for the prediction might also provide new information for the biological mechanism under the binding of TCR and HLA sequences.

## Conclusion

We proposed a neural network method named DePTH to predict TCR-HLA associations based on their amino acid sequences. DePTH can make prediction for any TCR-HLA pairs provided sequence information is available. This property makes it possible to study rare HLAs, which is hard to do by examining co-occurrence patterns, due to the limited TCR repertoire data available.

To demonstrate the utilities of DePTH, we use the prediction scores of TCR-HLA associations given by DePTH to quantify functional similarities between HLA alleles, and further define between-individual HLA distances and individual-level heterozygosity metrics. We show that some of these metrics are significantly associated with survival outcomes of cancer patients.

## Supporting information

**S1 Appendix. Supplementary text, tables and figures.**
(PDF)

## Author Contributions

**Conceptualization:** Si Liu, Wei Sun.

**Data curation:** Si Liu, Philip Bradley, Wei Sun.

**Formal analysis:** Si Liu.

**Funding acquisition:** Wei Sun.

**Investigation:** Si Liu, Philip Bradley, Wei Sun.

**Methodology:** Si Liu, Philip Bradley, Wei Sun.

**Project administration:** Wei Sun.

**Resources:** Si Liu, Philip Bradley, Wei Sun.

**Software:** Si Liu.

**Supervision:** Philip Bradley, Wei Sun.

**Validation:** Si Liu.

**Visualization:** Si Liu.

**Writing – original draft:** Si Liu, Wei Sun.

**Writing – review & editing:** Si Liu, Philip Bradley, Wei Sun.

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
