## [Decision Letter · Decision Letter 0]

11 Aug 2023

Dear Dr. Sun,

Thank you very much for submitting your manuscript "Neural network models for sequence-based TCR and HLA association prediction" for consideration at PLOS Computational Biology.

As with all papers reviewed by the journal, your manuscript was reviewed by members of the editorial board and by several independent reviewers. In light of the reviews (below this email), we would like to invite the resubmission of a significantly-revised version that takes into account the reviewers' comments.

The three reviewers have raised lots of important comments and suggests which are expected to be addressed in a revised version of the manuscript.

We cannot make any decision about publication until we have seen the revised manuscript and your response to the reviewers' comments. Your revised manuscript is also likely to be sent to reviewers for further evaluation.

Sincerely,

Jinyan Li

Academic Editor

PLOS Computational Biology

Sushmita Roy

Section Editor

PLOS Computational Biology

The three reviewers have raised lots of important comments and suggests which are expected to be addressed in a revised version of the manuscript.

Reviewer's Responses to Questions

**Comments to the Authors:**

Reviewer #1: Liu et al. proposed to use neural network models to predict the association between T cell receptors (TCR) and human leukocyte antigens (HLA) with their amino acid sequences. After constructing the training data of positive and negative pairs, they selected the tuning parameters in the deep learning model to predict associations. Benchmarking shows better performance than the existing method CLAIRE. The proposed DePTH differs from CLAIRE by different training data, and DePTH allows unseen HLA alleles in training data since it directly inputs the sequences. In the application of immune checkpoint blockade, the authors showed that DePTH-predicted TCR-HLA associations could be useful as biomarkers associated with clinical outcomes. The paper is well-written, the analyses are comprehensive, and the scientific contribution is strong. I list some minor comments below that may further improve the manuscript.

"Table C: Performance of DePTH on leave-one-out experiments"

The AUC is above 0.64, specificity is above 0.92, but recall/sensitivity is below 0.31. This suggests a better cutoff other than 0.5 may be selected to balance specificity and sensitivity in leave-one-out experiments. This may help predict new TCRs and HLAs unseen during training.

“The AUC to distinguish these 54 TCR-HLA pairs from the negative pairs from Emerson test data is 0.87”

The proposed DePTH with Emerson data performs well for the solved TCR-pMHC-I structures. I wonder how CLAIRE or DePTH with McPAS data performs. This can be a potential comparison as well.

“The validation AUC became stable after ensemble size increased to around 20 (S1 Appendix Fig B).”

Based on Fig B, shuffles # 1 and 3 have stable AUCs around 20 models, but the other two shuffles have AUCs increasing even after 20 models, which suggests more models can be used in the ensemble if computation is not a concern.

Reviewer #2: This manuscript developed a neural network method named DePTH to predict TCR-HLA associations based on their amino acid sequences. The method outperforms the existing method for the similar task for less common HLA alleles. The manuscript shows that DePTH can be used to quantify the functional similarities of HLA alleles. These similarities are shown to be associated with the survival outcomes of cancer patients who received immune checkpoint blockade treatment.

Major:

1. The comparison with random forest is better combined with Section “DePTH models achieve AUC around 0.8 on test TCR-HLA pairs” to give readers a reference on how good an AUC around 0.8 is.

2. (a) Please explain this sentence more clearly: Line 223: “CLAIRE treats HLA as a categorical variable, which limits its ability to generalize to unseen HLAs”. For example, how does CLAIRE treat HLA as a categorical variable. (b) Based on this statement, CLAIRE would have very low predictive power when being used to predict the association in a different dataset. But the results seem to show that the performance of DePTH is similar to CLAIRE. What is the explanation?

3. In section “DePTH outperforms CLAIRE on most HLAs when trained on the same data”, please provide the frequencies for all 10 HLA-I alleles.

4. The section on HLA similarity metrics is not easy to follow. It would help if the rationale of this analyses is explained more clearly at the beginning of the section. In particular, what are the clinical implications of HLA similarities (both between individual and individual-level heterogeneity)? How does TCR-HLA pair relate to HLA similarities? Also when reporting the results, please explain how to interpret the results of this analysis and the clinical implication.

5. What are the covariates in Fig 4? Please provide a more complete description of the survival analyses.

6. Some parts of the results section involve a long description of the analysis procedure. It may be helpful to accentuate the main message by putting some description of the procedure in the method section and providing more rationales.

Reviewer #3: The authors propose a method called DePTH for predicting TCR-HLA paring that uses the sequence information of HLA alleles (contrasting the competing method using HLA as a categorical variable), thus allow the prediction to generalize to rare and unseen HLAs.

Using the Emerson data to train the model with selective positive associations (co-occurrences of TCR and HLA) and sampled negative pairs as controls. Specifically, they selected 6,423 associated TCR-HLA pairs out of 742,832,595 TCR-HLA pairs involving HLA-I alleles and 11,037 associated TCR-HLA pairs out of 1,136,096,910 TCR-HLA pairs involving HLA-II alleles. They show DepTH outperformed the competing methods in prediction accuracy and further validated using an independent dataset by Szeto et al. The method has the potential to study rare HLAs. They show the method has moderate prediction accuracy for unseen pairs.

The study could be improved if examples of clinical utility can be demonstrated more convincingly. For example, in Crowell dataset, is there any novel findings of predicted TCR-HLA pairs associated with patient survival beyond what was reported in the original paper?

**Have the authors made all data and (if applicable) computational code underlying the findings in their manuscript fully available?**

Reviewer #1: Yes

Reviewer #2: Yes

Reviewer #3: Yes

PLOS authors have the option to publish the peer review history of their article (what does this mean?). If published, this will include your full peer review and any attached files.

Reviewer #1: **Yes: **Jiebiao Wang

Reviewer #2: No

Reviewer #3: No
---

## [Decision Letter · Decision Letter 1]

6 Nov 2023

Dear Dr. Sun,

We are pleased to inform you that your manuscript 'Neural network models for sequence-based TCR and HLA association prediction' has been provisionally accepted for publication in PLOS Computational Biology.

Best regards,

Jinyan Li

Academic Editor

PLOS Computational Biology

Sushmita Roy

Section Editor

PLOS Computational Biology

Reviewer's Responses to Questions

**Comments to the Authors:**

Reviewer #1: My comments have been fully addressed. Thank you!

Reviewer #2: Authors have addressed all my comments satisfactorily

Reviewer #3: The authors have addressed my critique.

**Have the authors made all data and (if applicable) computational code underlying the findings in their manuscript fully available?**

Reviewer #1: None

Reviewer #2: Yes

Reviewer #3: Yes

PLOS authors have the option to publish the peer review history of their article (what does this mean?). If published, this will include your full peer review and any attached files.

Reviewer #1: No

Reviewer #2: No

Reviewer #3: No

---

## [Editor Report · Acceptance letter]

15 Nov 2023

PCOMPBIOL-D-23-00978R1 

Neural network models for sequence-based TCR and HLA association prediction

Dear Dr Sun,

I am pleased to inform you that your manuscript has been formally accepted for publication in PLOS Computational Biology. Your manuscript is now with our production department and you will be notified of the publication date in due course.

With kind regards,

Zsofi Zombor
